**La bibliografia di Giacomo Caputo e il ruolo di Wikidata per la sua valorizzazione**

Il fondo librario di Giacomo Caputo (1901-1992), di proprietà dell'Università di Firenze e conservato presso il Museo e Istituto Fiorentino di Preistoria "Paolo Graziosi", costituisce una risorsa preziosa per indagare la produzione scientifica e il contesto intellettuale di uno dei protagonisti dell'archeologia del Novecento. Il progetto di ricerca *"Il fondo archivistico e librario di Giacomo Caputo: archeologia e restauro architettonico in una biblioteca d'autore"* si propone anche di valorizzare la sua bibliografia, includendo contributi inediti o poco noti della sua produzione letteraria emersi durante il censimento del fondo: si tratta di interventi, recensioni, note metodologiche appartenenti alla cosiddetta "letteratura grigia", nascosti all'interno di riviste, periodici e bollettini, ma che concorrono alla ricostruzione della bibliografia completa dell'autore. Attraverso Wikidata, il progetto mira a trasformare la bibliografia di Caputo in un ecosistema dinamico grazie all'interazione dei dati con altri elementi rilevanti, favorendo l'accessibilità globale, l'interoperabilità e la *data visualization*, superando quindi i limiti delle bibliografie statiche. Questa strategia non solo potenzia il corpus bibliografico, ma permette anche di restituire al vasto pubblico dati verificati e interrogabili, alimentando nuove suggestioni e contribuendo a tracciare nuove strade per la ricerca. Il progetto si inserisce nel dibattito attuale sull'utilizzo di metodologie innovative per la valorizzazione di bibliografie e biblioteche d'autore, proponendo un modello replicabile per la gestione e la valorizzazione di queste importanti raccolte.