# OpenReview forum: "La bibliografia di Giacomo Caputo e il ruolo di Wikidata per la sua valorizzazione"
_wikimedia.it/Wikidata_and_Research/2025/Conference — WD&R LT_

### Official Review · ~Elena_Marangoni1 · 2025-01-07
**Wikidata for a personal library and bibliography**

**Originality:** 4
**Impact:** 3
**Confidence:** 4

**Review:**

The proposal is very interesting for the idea of a enriched and dynamic bibliography, made possible thanks to the interaction with other data in Wikidata. I think the most valuable aspect is that of integration and data visualization for the discovery of new knowledge about the author and his milieu.

**Compliance:**

4

**Scientific Quality:**

4

---

### Official Review · ~Silvia_Bruni1 · 2025-01-13
**Wikidata per la creazione di una bibliografia dinamica e la valorizzazione un  un fondo di letterature grigia**

**Originality:** 4
**Impact:** 4
**Confidence:** 4

**Review:**

La sperimentazione di Wikidata come strumento per la creazione di una bibliografia dinamica è un in argomento di grande interesse.  Il fatto che riguardi un fondo di letteratura grigia e che comprenda spogli, difficili da  reperibili, è un merito ulteriore del progetto, così come l’arricchimento e la valorizzazione dei dati riferiti all'autore e al suo contesto disciplinare di riferimento.

**Compliance:**

5

**Scientific Quality:**

4

---

### Decision · Program_Chairs · 2025-01-23

Accept (LT)